# Prevalence and associated risk factors for hepatitis B and C viruses among refugee populations living in Mahama, Rwanda: A cross-sectional study

**Innocent Kamali** [1]*, **Dale A. Barnhart**[1,2], **Jean d'Amour Ndahimana**[1], **Kassim Noor**[3], **Jeanne Mumporeze**[3], **Françoise Nyirahabihirwe**[1], **Jean de la Paix Gakuru**[1], **Tumusime Musafiri**[1], **Sandra Urusaro**[1], **Jean Damascene Makuza** [4,5], **Janvier Serumondo**[4], **Dina Denis Rwamuhinda**[6], **Maurice Nkundibiza**[7], **Fredrick Kateera**[1], **Gallican Rwibasira Nshogoza**[4], **Joel M. Mubiligi**[1]

**1** Partners In Health-Rwanda/Inshuti Mu Buzima, Rwinkwavu, Rwanda, **2** Harvard Medical School, Department of Global Health and Social Medicine, Boston, Massachusetts, United States of America, **3** United Nations High Commissioner for Refugees (UNHCR), Kigali, Rwanda, **4** Rwanda Biomedical Centre, HIV/AIDS, STIs and OBBI Division, Kigali, Rwanda, **5** University of British Columbia, School of Population and Public Health, Vancouver, British Columbia, Canada, **6** Save the Children International, Kigali, Rwanda, **7** Alight, Kigali, Rwanda

* ikamali@pih.org

**Data Availability Statement:** Our ethical approvals from the Rwanda National Ethics Committee require the data to be stored on a secure PIH/IMB

## Abstract

### Introduction

As part of the integration of refugees into Rwanda's national hepatitis C elimination agenda, a mass screening campaign for hepatitis B (HBV) and hepatitis C (HCV) was conducted among Burundian refugees living in Mahama Camp, Eastern Rwanda. This cross-sectional survey used data from the screening campaign to report on the epidemiology of viral hepatitis in this setting.

### Methods

Rapid diagnostic tests (RDTs) were used to screen for hepatitis B surface antigen (HBsAg) and hepatitis C antibody (anti-HCV) among people of ≥15years old. We calculated seroprevalence for HBsAg and anti-HCV by age and sex and also calculated age-and-sex adjusted risk ratios (ARR) for other possible risk factors.

### Results

Of the 26,498 screened refugees, 1,006 (3.8%) and 297 (1.1%) tested positive for HBsAg and Anti-HCV, respectively. HBsAg was more prevalent among men than women and most common among people 25–54 years old. Anti-HCV prevalence increased with age group with no difference between sexes. After adjusting for age and sex, having a household contact with HBsAg was associated with 1.59 times higher risk of having HBsAg (95% CI: 1.27, 1.99) and having a household contact with anti-HCV was associated with 3.66 times higher risk of Anti-HCV (95% CI: 2.26, 5.93). Self-reporting having HBV, HCV, liver disease, or

server located within Rwanda or on password-protected computers of study collaborators. To meet these requirements, we are requesting that researchers with a reasonable request for the data contact IMBRC. Because refugees reflect a vulnerable population, it is our team's policy to also obtain approvals from UNHCR and MINEMA prior to conducting research. The data underlying the results presented in this study are available on reasonable request and pending written approval from the Inshuti Mu Buzima Research Committee (imbrc@pih.org), UNHCR, and MINEMA.

**Funding:** The authors received no specific funds for this work. DAB is supported by the Harvard Medical School Global Health Equity Research Fellowship, funded by Jonathan M. Goldstein and Kaia Miller Goldstein. The funders had no role in study design, data collection and analysis, decision to publish, or preparation of the manuscript.

**Competing interests:** The authors have declared that no competing interests exist.

previously screened for HBV and HCV were significantly associated with both HBsAg and anti-HCV, but RDT-confirmed HBsAg and anti-HCV statuses were not associated with each other. Other risk factors for HBsAg included diabetes (ARR = 1.97, 95% CI: 1.08, 3.59) and family history of hepatitis B (ARR = 1.32, 95% CI: 1.11, 1.56) and for anti-HCV included heart disease (ARR = 1.91, 95% CI: 1.30, 2.80) and history of surgery (ARR = 1.70, 95% CI: 1.24, 2.32).

## Conclusion

Sero-prevalence and risks factors for hepatitis B and C among Burundian were comparable to that in the Rwandan general population. Contact tracing among household members of identified HBsAg and anti-HCV infected case may be an effective approach to targeted hepatitis screening given the high risk among self-reported cases. Expanded access to voluntary testing may be needed to improve access to hepatitis treatment and care in other refugee settings.

## Introduction

Hepatitis B (HBV) and C (HCV) infections are the leading causes of cirrhosis, hepatocellular carcinoma, and liver-related deaths globally [1]. Although effective curable treatments are increasingly available, most people remain unaware of their hepatitis status until symptoms appear [2]. Asia and Africa are the two continents most affected by viral hepatitis infections [3], with sub-Saharan Africa having an estimated 6.1% prevalence of HBsAg [4] and overall 2.9% prevalence for hepatitis C antibodies (anti-HCV) [5]. In Rwanda a recent population-based study revealed the prevalence of HBsAg to be 2.0% and the prevalence of anti-HCV to be 1.2% among people 15–64 years old [6]. In response to the viral hepatitis burden, Rwanda established a national hepatitis program in 2011 with the first viral hepatitis guidelines disseminated in 2015 [7]. These guidelines were followed by the launch of a five-year HCV elimination plan in 2018, which was associated with increasing access to free viral hepatitis screening and treatment services accessible for Rwandans [8].

Rwanda, like many other African countries, hosts a large number of refugee populations. However, refugees were not initially included in Rwanda's national hepatitis program. In Europe and United states, refugee populations have been reported to have elevated risk of viral hepatitis compared to permanent residents of their host country. This risk is often attributed to poor living conditions during migration and resettlement [9–11]. However, it is also plausible that the higher risk of hepatitis B and C among immigrants and refugees may be primarily associated with differences in the prevalence of hepatitis between their host country and their country of origin rather [12, 13].

In December 2019, the Rwandan Ministry of Health (MoH) approved inclusion of refugees as part of the national hepatitis elimination plan and this was followed by a mass screening and treatment campaign initiated in the Mahama refugee camp; the largest camp in Rwanda hosting over 60,000 Burundian refugees that was established in 2015. Both Burundi and Rwanda are classified as countries with an intermediate burden of viral hepatitis B and C [14, 15]. In Burundi, the HBsAg prevalence has been estimated at 4.6% [16] and the anti-HCV prevalence at 8.2% [17]. However, these estimates are from the 2002, and little is known about the extent to which these prevalences can be generalized to Burundian refugees hosted in

Rwanda today, as well as whether there are any specific risk factors that are associated with hepatitis among refugees. This study aimed to determine the prevalence and risk factors of HBV and HCV among refugee populations living in Mahama camp, Rwanda.

## Methods

### Study design

This cross-sectional study used data from a mass screening campaign among refugee populations residing in Mahama camp, which was conducted in February and March 2020.

### Screening program & participants

This screening program was led by Partners In Health/Inshuti Mu Buzima (PIH/IMB), an international non-government organization that supports health care implementation in three rural Rwandan districts in partnership with other key stakeholders: Ministry in charge of Emergency Management (MINEMA), MoH, United nations High Commissioner for Refugees (UNHCR), Save the children and Alight. In compliance to the Rwandan national viral hepatitis guidelines, a voluntary mass screening campaign for HBV and HCV was available to all refugees aged $\geq$ 15 years living in Mahama camp, in Kirehe district, Eastern Province. Prior to the screening campaign, 18 people in the camp had already been initiated on viral hepatitis treatment in the HIV clinic where they were enrolled, but hepatitis screening and treatment was not otherwise available.

Briefly, community awareness for the campaign was done through meetings with health implementing organization-partners of UNHCR, Save the Children, and Alight. Executive committees, the elected representative of the refugees, and local community health workers, mobilized the population for the screening. On the day but prior to being screened, group education, information and communication session was conducted and verbal consent for screening for all eligible participants age 15 and above was obtained. All participants completed a digital registration form where data on location, demographic and risk factors were collected. During screening, trained nurses and laboratory technicians collected capillary blood samples. Lab testing was performed using SD Bioline rapid diagnostic tests (RDTs) manufactured by Abbott Diagnostics Korea Inc, Giheung-gu, Korea, with sensitivity and specificity of >99% to detect anti-HCV and sensitivity of 96.7% and specificity of 98.9% to detect HBsAg [18]. For those who screened positive, a 4–5 ml venous blood sample for HBV Deoxyribonucleic Acid (DNA) and/or HCV Ribonucleic Acid (RNA) was collected in an Ethylene diamine tetra-acetic acid (EDTA) tubes for viral load testing. Collected blood samples for viral load testing were arranged, first in a tube rack, put in a cooler and transported the same day to Rwamagana provincial hospital located at 98 kilometers from the screening site for analysis. At the hospital, samples were centrifuged, plasma separated from other blood components immediately and stored in a fridge at a temperature between 2–8˚C. Samples that could not be analyzed within 72 hours were stored in a freezer at -20˚C temperature while waiting further processing. Analysis was performed using COBAS AmpliPrep/COBAS TaqMan HCV and HBV Test, V.2.0: Quantitative (Roche) with a lower limit of quantification of 15 IU/mL. All sample analyzes were performed within one month from the time of their reception at the testing hub.

### Data sources

All data collected during patient registration, screening, and viral load testing was recorded in a Research Electronic Data Capture (REDCap) database for use to facilitate patient follow-up. Prior to analysis, data was extracted from REDCap, duplicate entries were identified and

removed from the database, and household contacts with HBsAg and Anti-HCV were identified. Names, identifying numbers, and address were removed from the dataset prior to analysis.

Data collected during the campaign included HBsAg and Anti-HCV seroprevalence, demographics (age, sex, marital status, level of education, year of arrival in Mahama camp)and self-reported clinical characteristics. Marital status was categorized (into widowed, divorced, or separated versus never married or currently married or cohabitating), education level (no schooling, primary, secondary, or university), self-reported comorbidities on HIV, HBV and HCV, diabetes, heart diseases, chronic renal failure and cancer, and risk factors (previously diagnosis with liver disease, history of hepatitis in the family, surgery, traditional operation, unhygienic medical or household practices, multiple sex partners). Having a household contact for HBsAg and anti-HCV was defined as living in the same plot as another person identified as having HBsAg or anti-HCV during the screening.

## Statistical methods and data analysis

Data were summarized using proportions and 95% confidence intervals (95% CIs). Prevalence were reported overall and also stratified by age and sex. To identify risk factors associated with HBsAg and anti-HCV, we dichotomized categorical risk factors and conducted bivariate analysis to calculate crude risk ratios (RR) with 95% confidence intervals and age-and-sex adjusted risk ratios (ARR) using the Mantel-Haenszel method. Adjusted risk ratios were calculated using strata defined by sex and 20-year age intervals. We also assessed associations between having a detectable viral load for HBV DNA or HCV RNA and age, sex, and comorbidities among individuals who screened positive for HBsAg or Anti-HCV. As a sensitivity analysis, we also assessed the association between demographic characteristics and having a viral load >20,000 IU/mL, which is the threshold for automatically qualifying for HBV treatment in Rwanda, among patients with detectable HBV DNA. To compare the prevalence among refugees in Mahama with, the recent results from the nationally-representative Rwanda Population-Based HIV Impact Assessment (RPHIA), which published prevalence of hepatitis B and C among Rwandans aged 15–64, we recalculated the crude prevalence of HBsAg and anti-HCV among 15 to 64 year-old participants in the Mahama screening program among men and women. To create directly comparable estimates that were not confounded by age, we also calculated prevalence that were standardized to the age distribution of RPHIA survey respondents using 5-year age categories. Finally, we reported the indirectly standardized prevalence ratios (SPR) and 95% confidence intervals comparing the observed number of Mahama residents who screened positive for HBsAg or anti-HCV with the number of positive cases that we would have expected if each age group in Mahama had experienced the same age-specific prevalence as reported in the RPHIA report. The interpretation of the SPR is similar to the interpretation of the more commonly used Standardized Mortality Ratio, in that a SPR greater than 1 would reflect that, after adjusting for age, Mahama residents have a higher prevalence than Rwandan general population while an SPR less than 1 reflects a lower prevalence among Mahama residents relative to the general population. All analyses were conducted using STATA version 15.1 [19].

## Ethics statement

The study was approved by Inshuti Mu Buzima Research Committee (IMBRC) and Rwanda National Ethics Committee (RNEC) 015/RNEC/2020). Clinicians received oral consent from all participants aged 15 and older during the screening process. However, this study used retrospective data, which was collected as part of routine clinical practice; therefore, informed

consent was not required by RNEC. All methods were performed in accordance with local guidelines and regulations. The screening campaign was also approved by the Ministry of Health through its implementing agency Rwanda Biomedical Centre (RBC).

## Results

### Participant's characteristics

Socio-demographic and self-reported clinical comorbidities of the 26,498 unique individuals who participated in the screening campaign are presented in Table 1. More than half of the campaign participants (53.8%) were female. The majority of participants were married or lived with a partner (58.4%). About one third (33.4%) had no formal education and another third had some secondary school or higher (32.1%). Almost all refugees reported to have directly come to Rwanda after leaving their country and most of them (77.8%) arrived in 2015. The most commonly reported co-morbidities were HIV (3.3%) and heart diseases (2.7%), while the most common self-reported risk factors were unhygienic medical or household practices 12,198 (46%), multiple sex partners 8,839 (33.4%), and traditional operation 8,246 (31.1%).

### Prevalence by age and sex

Of the 26,498 individuals who were RDT screened, 1,006 (3.8% (95% CI 3.57, 4.03), were found HBsAg positive, with higher prevalence among men in all age categories and highest among men aged 25–54 years old (Fig 1). A similar age pattern with highest prevalence among middle-aged individuals was observed among women. Of the 26,498 people screened, those with positive anti-HCV were 297(1.1%) (95% CI: 1.00, 1.25). In both sexes anti-HCV positivity increased with age with no significant differences between the sexes (Fig 2). Only nine cases had HBV and HCV co-infections.

### Risk factors

Table 2 shows crude age- and sex-adjusted risk ratios for possible risk factors of HBsAg and anti-HCV. After adjusting for the distribution age and sex, we found that no additional demographic characteristics were associated with either HBsAg or anti-HCV positivity. Among self-reported comorbidities, diabetes (ARR = 1.97, 95% CI: 1.08, 3.59) was associated with HBsAg positivity while heart disease was associated with anti-HCV positivity [ARR = 1.91, 95% CI: 1.30, 1.80]. Patients who self-reported having HBV, HCV, liver disease, or having previously screened for HBV and HCV were significantly more likely to have HBsAg as well as anti-HCV. However, those with RDT-confirmed HBsAg were not more likely to be positive for anti-HCV (ARR = 1.05 95% CI: 0.59, 1.87) and those with RDT-confirmed anti-HCV were not more likely to be positive for HBsAg (ARR = 1.05, 95% CI: 0.58, 1.91). Having a household contact with HBsAg was significantly associated with having HBsAg (ARR = 1.32, 95%: 1.12,1.56) while having a household contact with anti-HCV was significantly associated with anti-HCV (ARR = 3.66, 95% CI: 2.26, 5.92). Other risk factors for HBV include having history of hepatitis in the family (ARR = 1.32, 95% CI:1.12, 1.56) while other risk factors for HCV included history of surgery (ARR = 1.69,95% CI: 1.24, 2.31).

The prevalence of HBsAg and anti-HCV did not meaningfully change when we restricted our sample to residents aged 15–64 or when we standardized our age distribution to the ages of respondents to the Rwandan Population-based HIV Impact Assessment 2018–2019 (Table 3). Our indirectly standardized prevalence ratios suggest that after adjusting for age, Mahama residents have a 1.86 times higher prevalence of HBsAg than the Rwandan general population (95% CI: 1.74–1.98). The prevalence of HBsAg is, especially elevated among

**Table 1. Sociodemographic and self-reported clinical characteristics.**

| Variable | Frequency | % |
|---|---|---|
| **Sex (N = 26,285)** | | |
| Female | 14,136 | 53.8 |
| Male | 12,149 | 46.2 |
| **Age (N = 26,323)** | | |
| 15–24 | 7,159 | 27.2 |
| 25–34 | 8,632 | 32.8 |
| 35–44 | 5,267 | 20.0 |
| 45–54 | 2,778 | 10.6 |
| 55–64 | 1,482 | 5.6 |
| 65+ | 1,005 | 3.8 |
| **Marital Status (25,829)** | | |
| Never married | 7,265 | 28.1 |
| Married or living together | 15,095 | 58.4 |
| Widowed | 1,569 | 6.1 |
| Divorced | 271 | 1 |
| Separated | 1,629 | 6.3 |
| **Highest Education Level (N = 26,417)** | | |
| No School | 8,836 | 33.4 |
| Primary | 9,084 | 34.3 |
| Secondary | 7,990 | 30.2 |
| University | 507 | 1.9 |
| **Year of Arrival in Mahama camp (N = 26,392)** | | |
| 2015 | 20,564 | 77.9 |
| 2016 | 2,731 | 10.3 |
| 2017 | 1,417 | 5.4 |
| 2018 | 880 | 3.3 |
| 2019 | 646 | 2.4 |
| 2020 | 154 | 0.6 |
| **Co-morbidities[1] (N = 25,442)** | | |
| HIV | 850 | 3.3 |
| Heart diseases | 675 | 2.7 |
| Chronic renal failure | 264 | 1.0 |
| Diabetes | 140 | 0.6 |
| Self-reported HBV | 118 | 0.5 |
| Self-reported-HCV | 76 | 0.3 |
| Cancer | 28 | 0.1 |
| **Risk factors[1] (N = 26,498)** | | |
| Unhygienic medical or household practices | 12,298 | 46.0 |
| Multiple sexual partners | 8,839 | 33.4 |
| Physical trauma | 8,260 | 31.2 |
| Had a traditional operation | 8,246 | 31.1 |
| Had hepatitis in the family | 3,206 | 12.1 |
| Surgery | 2,777 | 10.5 |
| Diagnosed with liver disease | 1,054 | 4.0 |
| Previously screened for HBV/HCV | 780 | 2.9 |
| **Household contact (N = 26,317)** | | |
| Household contact with HBsAg | 1,366 | 5.2 |

(*Continued*)

**Table 1.** (Continued)

| Variable | Frequency | % |
|---|---|---|
| Household contact with Anti-HCV | 401 | 1.5 |

[1]Individuals could report more than one comorbidity and risk factor.

women (SPR = 2.44, 95% CI: 2.21–2.69). The prevalence of anti-HCV is significantly lower in the overall Mahama population than among the Rwandan population (SPR = 0.84, 95% CI: 0.74–0.96), driven largely by a lower prevalence among men (SPR = 0.71, 95% CI: 0.58–0.87).

## Viral load test results by characteristics

Table 4 shows that, of the 1006 patients who screened positive for HBsAg, 916 (91.0%) had viral load results returned and of these 916, 781 (85.3%, 95% CI: 82.80, 87.50) had detectable HBV DNA. Male sex and younger age were both associated with viral load positivity for HBV as well as with having a viral load >20,000 IU/mL among those with detectable viral loads (S1 Appendix). For HCV, of 297 of patients who screened positive, 271 (91.2%) had valid viral load results returned of which 213 (78%, 95% CI: 73.23, 83.33) had detectable HCV RNA positive results.

## Discussion

This paper is the first to measure HBsAg and anti-HCV prevalence and assess for associated risk factors among refugee populations living in Rwanda. The 3.8% prevalence of HBsAg

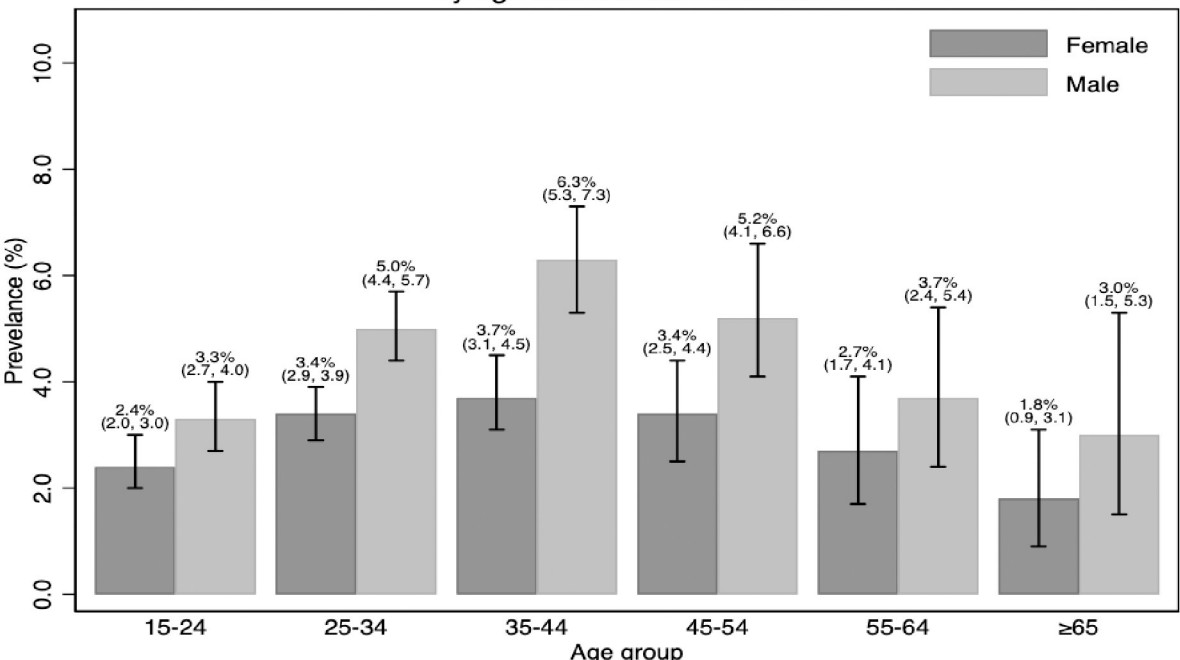

**Fig 1. Hepatitis B surface antigen prevalence by age and sex.**

**Fig 2. Hepatitis C antibody prevalence by age and sex.**

shown in our study is slightly higher than what has been reported in Rwanda's recent population-based survey (2.0%) but comparable to previous estimate from mass screening campaigns among the Rwandan general population (3.9%). This elevated risk among refugees persisted even after we standardized our age distribution to the age of respondents to the RPHIA. Overall, the observed prevalence of HBsAg was almost twice higher among Mahama residents than we would have expected if Mahama residents had exhibited the same age-specific HBsAg prevalence's as the respondents to the RPHIA survey. As elsewhere in Rwanda, we also found that men are at higher risk than women and that prevalence is highest among middle-aged individuals [16]. Anti-HCV prevalence among refugees (1.1%) was similar to the prevalence among Rwandans who participated in the RPHIA (1.2%). However, the observed prevalence of anti-HCV was slightly less among Mahama residents than we would have expected if Mahama residents had exhibited the same age-specific prevalence's as the respondents to the RPHIA survey. This lower prevalence of anti-HCV among Mahama residents may largely reflect reduced HCV prevalences at older ages. We observed that 2.8% and 3.1% of male and female Mahama residents aged 55–64 were anti-HCV positive, respectively. However, RPHIA reported prevalences of between 4.3% and 11.0% for male and female Rwandans age between 50–64 [6].

HCV prevalence among refugees was much lower compared to prevalences from the previously campaign among people aged 25 years and above nationally (6.8%) and in Kirehe district, where Mahama refugee camp is located, (8.4%) [20]. Similar to what has been reported in previous campaigns in Rwanda, we found that anti-HCV prevalence increases with age with no differences between the sexes. This increased risk with age may be explained by historic unhygienic medical procedures, especially for those conducted by traditional practitioners, before the establishment of infection control policies. However, the risk of anti-HCV among Rwandans 50–64 was also much higher compared to among refugees of the same age.

**Table 2. Risk ratios for the association between risk factors and hepatitis B surface antigen or hepatitis C antibody.**

| Risk Factor | Hepatitis B Surface Antigen | | | Hepatitis C Antibody | | |
|---|---|---|---|---|---|---|
| | N | Crude Risk Ratio | Adjusted Risk Ratio | N | Crude Risk Ratio | Adjusted Risk Ratio |
| **Socio-demographics** | | | | | | |
| Widowed, separated, or divorced | 26,045 | 0.84 (0.77, 10) | 0.94 (0.77, 1.15) | 26,058 | 2.87 (2.26, 3.63) | 1.32 (0.99, 1.76) |
| No Schooling | 26,045 | 0.96 (0.84, 1.09) | 1.04 (0.91, 1.20) | 26,058 | 1.85 (1.48, 2.32) | 0.86 (0.67, 1.12) |
| **Self-reported comorbidities** | | | | | | |
| HIV | 24,999 | 0.91 (0.63, 1.31) | 0.92 (0.63, 1.32) | 25,012 | 1.1 (0.59, 2.06) | 0.86 (0.45, 1.61) |
| Diabetes | 24,999 | 1.91 (1.05, 3.49) | 1.97 (1.08, 3.59) | 25,012 | 4.05 (1.83, 8.94) | 2.14 (0.97, 4.71) |
| Heart diseases | 24,999 | 1.07 (0.74, 1.56) | 1.19 (0.81, 1.74) | 25,012 | 4.35 (2.99, 6.35) | 1.91 (1.30, 2.80) |
| Chronic renal failure | 24,999 | 1.53 (0.93, 2.51) | 1.59 (0.97, 2.62) | 25,012 | 1.78 (0.74, 4.28) | 0.97 (0.4, 2.33) |
| Cancer | 24,999 | 0.95 (0.14, 6.52) | 0.93 (0.14, 6.26) | 25,012 | 3.33 (0.48, 22.88) | 2.56 (0.41, 16.12) |
| **Self-reported hepatitis status** | | | | | | |
| Hepatitis B | 24,999 | 12.61 (10.22, 15.55) | 11.81 (9.54, 14.63) | 25,012 | '--- | '--- |
| Hepatitis C | 24,999 | 2.82 (1.46, 5.44) | 2.68 (1.38, 5.19) | 25,012 | 20.72 (13.18, 32.56) | 17.44 (11.25, 27.04) |
| Diagnosed with liver disease | 26,045 | 2.46 (2.01, 3.02) | 2.41 (1.96, 2.96) | 26,058 | 2.32 (1.56, 3.45) | 2.23 (1.51, 3.32) |
| Previously screened for HBV/HCV | 26,045 | 2.8 (2.25, 3.48) | 2.69 (2.16, 3.35) | 26,058 | 3.56 (2.44, 5.18) | 3.15 (2.17, 4.57) |
| **RDT confirmed HCV/HBV status** | | | | | | |
| Screened positive for HBV | 26,045 | '--- | '--- | 26,018 | 0.97 (0.53, 1.77) | 1.05 (0.58, 1.91) |
| Screened positive for HCV | 26,018 | 0.97 (0.54, 1.75) | 1.05 (0.59, 1.87) | 26,058 | '--- | '--- |
| **Household contacts** | | | | | | |
| Household contact HBV RDT+ | 25,898 | 1.59 (1.27, 1.98) | 1.59 (1.27, 1.99) | 25,910 | '--- | '--- |
| Household contact HCV RDT+ | 25,898 | '--- | '--- | 25,910 | 3.67 (2.24, 6.01) | 3.66 (2.26, 5.93) |
| **Other risk factors** | | | | | | |
| Had hepatitis in the family | 26,045 | 1.32 (1.12, 1.57) | 1.32 (1.11, 1.56) | 26,058 | 1.33 (0.98, 1.82) | 1.21 (0.89, 1.65) |
| Surgery | 26,045 | 0.98 (0.81, 1.20) | 1.02 (0.84, 1.25) | 26,058 | 1.56 (1.15, 2.14) | 1.7 (1.24, 2.32) |
| Had a traditional operation | 26,045 | 1.02 (0.89, 1.16) | 1.02 (0.89, 1.16) | 26,058 | 1.34 (1.06, 1.69) | 0.95 (0.75, 1.20) |
| Multiple sex partners | 26,045 | 1.18 (1.04, 1.33) | 1.05 (0.93, 1.20) | 26,058 | 0.77 (0.6, 1) | 0.88 (0.68, 1.14) |
| Physical trauma | 26,045 | 0.93 (0.81, 1.06) | 0.89 (0.78, 1.02) | 26,058 | 0.81 (0.63, 1.05) | 0.79 (0.61, 1.02) |
| Unhygienic medical or household practices | 26,045 | 0.96 (0.85, 1.08) | 0.97 (0.86, 1.10) | 26,058 | 0.84 (0.67, 1.06) | 0.82 (0.66, 1.04) |
| Arrived in Mahama after 2015 | 25799 | 1.29 (0.85, 1.95) | 1.22 (0.81, 1.84) | 25812 | 1.61 (0.8, 3.23) | 1.65 (0.83, 3.29) |

**Table 3. Prevalence of HBsAg and anti-HCV among Mahama residents aged 15–64 compared to age-standardized prevalence in the general population.**

| Infection by sex | Crude prevalence and 95% CI among Mahama residents 15–64 | Age-standardized prevalence and 95% CI among Mahama residents 15–64 | Indirectly standardized prevalence ratio and 95% CI[1] |
|---|---|---|---|
| **HBsAg** | | | |
| Female | 3.1% (2.8%, 3.4%) | 3% (2.7%, 3.3%) | 2.44% (2.21, 2.69) |
| Male | 4.7% (4.4%, 5.1%) | 4.5% (4.2%, 4.9%) | 1.58% (1.46, 1.72) |
| Total | 3.9% (3.6%, 4.1%) | 3.7% (3.5%, 4%) | 1.86% (1.74, 1.98) |
| **Anti-HCV** | | | |
| Female | 1% (0.8%, 1.2%) | 1.1% (0.9%, 1.3%) | 0.99% (0.83, 1.17) |
| Male | 0.8% (0.7%, 1%) | 0.9% (0.7%, 1.1%) | 0.71% (0.58, 0.87) |
| Total | 0.9% (0.8%, 1%) | 1% (0.9%, 1.1%) | 0.84% (0.74, 0.96) |

[1]Data age distribution and age- and sex-specific hepatitis prevalence extracted from the Rwandan Population-based HIV Impact Assessment 2018–2019 [6].

Although our campaign in Mahama had extremely high coverage (76.9%) and is likely to be a relatively accurate estimate of the prevalence of HBV and HCV among refugees, we would expect a voluntary screening campaign to result in higher prevalence than a population-based survey as individuals who have prior knowledge or strong reasons to suspect that they may have hepatitis will be more motivated to participate in the campaign. This mechanism could also explain why the previous estimates from Rwanda's general population mass screening campaigns were higher than those from their recent population-based survey.

Our estimates for the prevalence of HCV among Burundian refugees are much lower than what was previously reported in a 2002 population-based survey in Burundi (8.2%) [17], but is similar to the HCV prevalence among Burundian blood donors (1.04%) [21]. Research in Italy [13] and six other western countries [22] have found that the prevalence of hepatitis B, among refugees was more comparable to the prevalence of their country of origin than their host countries. However, it is also possible that the prevalence among refugees is lower than individuals in the host country due to the "healthy migrant effect, where migrants tend to be healthier than the general population of both their home country and receiving countries [23–26]. Notably, we did not observe any association between duration of stay in Mahama and risk of HBV or HVC, suggesting that transmission within the camp and between the refugees and the Rwandan host population was likely rare during our study period. However, it is important to note that this study is cross-sectional study and therefore cannot provide conclusive evidence against transmission within the camp.

As expected, patients who self-reported as having HBV were more likely to have RDT-confirmed HBsAg and those who self-reported as having HCV were more likely to have RDT-confirmed anti-HCV. However, self-reported HCV status was also associated with HBsAg, even though we did not observe an association between having RDT-confirmed HBsAg and RDT-confirmed anti-HCV (Table 4). We believe these finding largely reflects confusion among patients regarding which type of hepatitis they have been diagnosed with, and highlights the importance of high-quality patient education and confirmatory testing. These finding also suggests that even in the absence of mass screening campaigns, refugees with hepatitis may independently present for hepatitis treatment at health facilities if resources are made available to them.

As it has been previously reported in Rwanda and Egypt, we also found possible evidence of household transmission of both HBV and HCV [27, 28]. Within-household transmission could reflect a combination of sexual transmission between partners and vertical transmission during pregnancy, as has been documented for both HBV and HCV [29]. Additionally, people living in the same house or plot may be more likely to share sharp materials such as razors, scissors, and other personal hygiene items, which could transmit hepatitis. As previously suggested, screening of household contacts could be an efficient way to of identifying additional people with HBV or/and HCV who can benefit from treatment and follow-up [30]. Additional research may be needed to assess the feasibility of this strategy and to identify whether screening of all household contacts or only of sexual partners is most efficient.

Our findings on viral load positivity reflect a similarity with previous study conducted in Rwanda where 83% of the HBsAg positive had detectable HBV DNA while 72.2% of anti-HCV positive were confirmed with HCV RNA [27]. Younger men were not only more likely to be HBsAg positive, they were also more likely to have detectable viral load.

This study had a number of strengths. First, the screening campaign covered 76.9% of the target population and over 90% for women aged 30-64years old. This high coverage of the screening campaign gives us confidence that the population who participated in our screening campaign is reasonably representative of the general population living in Mahama. Second, we had low levels of missing data. Finally, due to the highly structured addresses

**Table 4. Viral load results by sociodemographic and clinical characteristics.**

| Variable | HBV DNA Test result | | p-value | HCV RNA Test result | | p-value |
|---|---|---|---|---|---|---|
| | Undetectable (<20 IU/mL) | Detectable (≥ 20 IU/mL) | | Undetectable (< 15 IU/mL) | Detectable (≥15 IU/mL) | |
| | N = 135 | N = 781 | | N = 58 | N = 213 | |
| **Sex** | | | 0.003 | | | 0.13 |
| Female | 73 (54.5%) | 315 (40.6%) | | 40 (69.0%) | 121 (57.3%) | |
| Male | 61 (45.5%) | 461 (59.4%) | | 18 (31.0%) | 90 (42.7%) | |
| **Age** | | | 0.004 | | | 0.85 |
| 15–24 | 21 (15.6%) | 162 (20.8%) | | 4 (6.9%) | 8 (3.8%) | |
| 25–34 | 36 (26.7%) | 292 (37.6%) | | 12 (20.7%) | 47 (22.1%) | |
| 35–44 | 37 (27.4%) | 191 (24.6%) | | 13 (22.4%) | 42 (19.7%) | |
| 45–54 | 25 (18.5%) | 84 (10.8%) | | 9 (15.5%) | 37 (17.4%) | |
| 55–64 | 12 (8.9%) | 33 (4.2%) | | 9 (15.5%) | 29 (13.6%) | |
| 65+ | 4 (3.0%) | 15 (1.9%) | | 11 (19.0%) | 50 (23.5%) | |
| **Self-reported co-morbidities/Risk factors** | | | | | | |
| Hepatitis B | 12 (9.0%) | 37 (5.0%) | 0.07 | | | |
| Hepatitis C | 1 (0.8%) | 6 (0.8%) | 1.00 | 5 (8.9%) | 9 (4.5%) | 0.20 |
| Diabetes | 2 (1.5%) | 8 (1.1%) | 0.65 | 1 (1.8%) | 5 (2.5%) | 1.00 |
| Heart diseases | 5 (3.8%) | 19 (2.6%) | 0.39 | 5 (8.9%) | 21 (10.6%) | 0.81 |
| Chronic renal failure | 5 (3.8%) | 10 (1.3%) | 0.06 | 3 (5.4%) | 2 (1.0%) | 0.07 |
| Cancer | 0 (0.0%) | 1 (0.1%) | 1.00 | 0 (0.0%) | 1 (0.5%) | 1.00 |
| Household contact HBV RDT+ | | | 0.87 | | | 0.35 |
| No | 124 (92.5%) | 713 (91.8%) | | 53 (91.4%) | 202 (94.8%) | |
| Yes | 10 (7.5%) | 64 (8.2%) | | 5 (8.6%) | 11 (5.2%) | |
| Household contact HCV RDT+ | | | | | | 0.35 |
| No | | | | 53 (91.4%) | 202 (94.8%) | |
| Yes | | | | 5 (8.6%) | 11 (5.2%) | |

within Mahama, we were able to estimate risk among household contacts, which would not have been possible in a general population setting. In addition, we were able to assess risk factors for both seroprevalence and hepatitis chronicity. However, there are also some limitations. First, we used a clinical data source, and some missing data, especially on VL results, did occur. Second, risk factors were self-reported and responses may have been affected by social desirability bias. Third, our screening strategy relied on HBsAg, and therefore some cases of occult hepatitis may have been escaped detection. Finally, the prevalence estimates from our study may not be generalizable to other refugee or displaced populations, particularly refugee populations coming from countries of origin with different prevalence of hepatitis B and C.

## Conclusions

HBsAg and anti- HCV sero-prevalence among Burundian refugees living in Rwanda were comparable to those of Rwanda general population. Our results suggest additional research into targeted screening among household contacts of clients who screen positive for hepatitis B or hepatitis C as an efficient strategy may be warranted to identify additional viral infected index clients who could benefit from hepatitis screening and treatment.

## Supporting information

**S1 Appendix. Association between demographic & clinical characteristics and having a viral load >20,000 IU/mL.**
(DOCX)

## Acknowledgments

This study would not have been possible without the contribution of refugees living in Mahama camp, Rwanda who during this campaign provided the information used in this study. We would additionally like to thank the hepatitis-screening program, MINEMA, MoH, RBC, UNHCR, Alight, and Save the Children, the refugee executive committee, and the Mahama Community Health Workers for their support in this process.

## Author Contributions

**Conceptualization:** Innocent Kamali, Jean d'Amour Ndahimana, Jeanne Mumporeze, Jean de la Paix Gakuru, Fredrick Kateera.

**Data curation:** Dale A. Barnhart.

**Formal analysis:** Innocent Kamali, Dale A. Barnhart.

**Methodology:** Dale A. Barnhart.

**Project administration:** Françoise Nyirahabihirwe.

**Supervision:** Innocent Kamali, Jean d'Amour Ndahimana, Kassim Noor, Jeanne Mumporeze, Françoise Nyirahabihirwe, Jean de la Paix Gakuru, Tumusime Musafiri, Dina Denis Rwamuhinda, Maurice Nkundibiza.

**Validation:** Jean d'Amour Ndahimana, Françoise Nyirahabihirwe, Jean de la Paix Gakuru, Tumusime Musafiri, Janvier Serumondo, Dina Denis Rwamuhinda, Fredrick Kateera, Gallican Rwibasira Nshogoza, Joel M. Mubiligi.

**Writing – original draft:** Innocent Kamali.

**Writing – review & editing:** Dale A. Barnhart, Jean d'Amour Ndahimana, Kassim Noor, Jeanne Mumporeze, Françoise Nyirahabihirwe, Jean de la Paix Gakuru, Tumusime Musafiri, Sandra Urusaro, Jean Damascene Makuza, Janvier Serumondo, Dina Denis Rwamuhinda, Maurice Nkundibiza, Fredrick Kateera, Gallican Rwibasira Nshogoza, Joel M. Mubiligi.

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
