## [Decision Letter · Decision Letter 0]

29 Jul 2021

PONE-D-21-14910

Prevalence and associated risk factors for hepatitis B and C viruses among refugee populations living in Mahama, Rwanda: A cross-sectional study

PLOS ONE

Dear Dr. Kamali,

Thank you for submitting your manuscript to PLOS ONE. After careful consideration, we feel that it has merit but does not fully meet PLOS ONE’s publication criteria as it currently stands. Therefore, we invite you to submit a revised version of the manuscript that addresses the several important points raised during the review process.

We look forward to receiving your revised manuscript.

Kind regards,

Isabelle Chemin, PhD

Academic Editor

PLOS ONE

Journal Requirements:

“DAB is supported by the Harvard Medical School Global Health Equity Research Fellowship, funded by Jonathan M. Goldstein and Kaia Miller Goldstein.”

“The authors received no specific funds for this work. DAB is supported by the Harvard Medical School Global Health Equity Research Fellowship, funded by Jonathan M. Goldstein and Kaia Miller Goldstein.

The funders had no role in study design, data collection and analysis, decision to publish, or preparation of the manuscript”

Reviewers' comments:

Reviewer's Responses to Questions

**Comments to the Author**

1. Is the manuscript technically sound, and do the data support the conclusions?

Reviewer #1: Yes

2. Has the statistical analysis been performed appropriately and rigorously? 

Reviewer #1: Yes

3. Have the authors made all data underlying the findings in their manuscript fully available?

Reviewer #1: Yes

4. Is the manuscript presented in an intelligible fashion and written in standard English?

Reviewer #1: Yes

5. Review Comments to the Author

Reviewer #1: In this cross-sectional study carried out from February to March 2020, the authors sought to determine the prevalence of hepatitis B and C and the associated risk factors among Burundian refugees living in Mahama camp in Rwanda. A total of 26,498 refugees aged 15 years and over were screened with rapid diagnosis test and the prevalence of HBsAg and anti-HCV were 3.8% and 1.1%, respectively. HBV-DNA and HCV-RNA quantification were performed in positive HBsAg and anti-HCV Burundian refugees. Other HBV markers haven’t been tested. Associated risk factors observed were family history (RRadjusted=1.32, 95% CI: 1.11, 1.56) and diabetes (RRadjusted=1.97, 95% CI: 1.08, 3.59) for HBsAg and history of surgery (RRadjusted=1.70, 95% CI: 1.24, 2.32) and heart disease (RRadjusted=1.91, 95% CI: 1.30, 2.80) for HCV antibodies.

The lines of the text haven’t been numbered.

Methods

The methods used to quantify HBV DNA and HCV-ARN must be specified as well as their detection and quantification limits. The authors should also specify the sensitivity and specificity of the rapid diagnostic tests used.

Since HCV is an RNA virus, it is very sensitive to temperature. Then condition of transport and storage (temperature) before processing viral load need to be indicated.

Did Refugees aged over 15–18 years give their assent and parents or tutors their consents?

Results

It would be interesting to have the rate of HBV-HCV co-infection among refugees in the Mahama camp since tests had been performed in the same population.

Table 3 could be restricted to the prevalence per age range observed on the current study in Mahama camp while the comparative analysis using the already published RPHIA data could be included in the discussion.

Table 4

Were patients with viral load at the detection thresholds (20 IU for HBV DNA and 15 IU for HCV RNA) considered as detectable or undetectable? Revise one of the signs of equality (table 4) to avoid confusion.

If data are available, it would have been interesting to see proportion of refugees who replicated the most (high viral load e.g., for HBV). Stratifying viral load will help to better orient strategies to limit contamination in the camp.

Discussion

Paragraph 2:

“We did not observe any association between duration of stay ……transmission within the camp and between the refugees and Rwandan host population is likely rare”

Even if we know that the main transmission route for HBV in Africa is vertical (mother-to-child) and horizontal (during childhood) transmission, it is difficult to conclude that transmission in the camp is rare because this current publication is a cross-sectional study. In addition, other HBV markers have not been evaluated.

Limits:

Since other HBV markers have not been tested (e.g., anti-HBc antibodies), occult hepatitis may escape to screening.

References

Revise references 6 and 19 (they are not accessible)

6. PLOS authors have the option to publish the peer review history of their article (what does this mean?). If published, this will include your full peer review and any attached files.

Reviewer #1: No

---

## [Author Response · Author response to Decision Letter 0]

17 Aug 2021

“Prevalence and associated risk factors for hepatitis B and C viruses among refugee populations living in Mahama, Rwanda: A cross sectional study"

PLOS ONE Ref. PONE-D-21-14910

We thank the reviewers for their positive comments and for the opportunity to revise and strengthen this manuscript. We have responded to all comments, as outlined below. Reviewer comments are in bold italics text, our response is in normal text with additions in bold and removed text shown with a strikethrough. 

Reviewer’s comment #1: 

Reviewer #1: In this cross-sectional study carried out from February to March 2020, the authors sought to determine the prevalence of hepatitis B and C and the associated risk factors among Burundian refugees living in Mahama camp in Rwanda. A total of 26, 498 refugees aged 15 years and over were screened with rapid diagnosis test and the prevalence of HBsAg and anti-HCV were 3.8% and 1.1%, respectively. HBV-DNA and HCV-RNA quantification were performed in positive HBsAg and anti-HCV Burundian refugees. Other HBV markers haven’t been tested. Associated risk factors observed were family history (RRadjusted=1.32, 95% CI: 1.11, 1.56) and diabetes (RRadjusted=1.97, 95% CI: 1.08, 3.59) for HBsAg and history of surgery (RRadjusted=1.70, 95% CI: 1.24, 2.32) and heart disease (RRadjusted=1.91, 95% CI: 1.30, 2.80) for HCV antibodies.

The lines of the text haven’t been numbered.

The lines of the text (the whole manuscript) are now numbered and double spaced.

Reviewer Comment #2 

Methods

The methods used to quantify HBV DNA and HCV-ARN must be specified as well as their detection and quantification limits. The authors should also specify the sensitivity and specificity of the rapid diagnostic tests used.

Comment addressed under Methods section, on page 6; lines 108-111

During screening, trained nurses and laboratory technicians collected capillary blood samples. Lab testing was performed using SD Bioline rapid diagnostic tests (RDTs) manufactured by Abbott Diagnostics Korea Inc, Giheung-gu, Korea, with sensitivity and specificity of ˃99% to detect anti-HCV and sensitivity of 96.7% and specificity of 98.9% to detect HBsAg(18)

Since HCV is an RNA virus, it is very sensitive to temperature. Then condition of transport and storage (temperature) before processing viral load need to be indicated.

Comment addressed on pages 6-7, lines 114-122

Collected blood samples for viral load testing were arranged in a tube rack, stored in a cooler, and transported the same day to Rwamagana provincial hospital, located 98 kilometers from the screening site, for viral load testing. At the hospital, samples were centrifuged, plasma separated from other blood components immediately and stored in a fridge at a temperature between 2-8oC. Samples that could not be analyzed with 72 hours were stored in a freezer at -20oC temperature while waiting further processing. Analysis was performed using COBAS AmpliPrep/COBAS TaqMan HCV and HBV Test, V.3.0: Quantitative (Roche) with a lower limit of quantification of 15 IU/mL. All sample analyzes were performed within one month from the time of their reception at the testing hub.

Did Refugees aged over 15–18 years give their assent and parents or tutors their consents?

Comment addressed under Ethics statement , on page 9, lines 170-175 

We have added a sentence on Ethics statement section, and it reads as follows:

The study was approved by Inshuti Mu Buzima Research Committee (IMBRC) and Rwanda National Ethics Committee (RNEC) 015/RNEC/2020).Clinicians received oral consent from all participants aged 15 and older during the screening process. However, this study used retrospective data, which was collected as part of routine clinical practice; therefore, informed consent was not required by RNEC. All methods were performed in accordance with local guidelines and regulations. The screening campaign was also approved by the Ministry of Health through its implementing agency Rwanda Biomedical Centre (RBC)

Results

It would be interesting to have the rate of HBV-HCV co-infection among refugees in the Mahama camp since tests had been performed in the same population.

Comment addressed on page 11, line 199

Of the 26,498 individuals who were RDT screened, 1,006 (3.8% (95% CI 3.57, 4.03), were found HBsAg positive, with higher prevalence among men in all age categories and highest among men aged 25-54 years old (Figure 1). A similar age pattern with highest prevalence among middle-aged individuals was observed among women. Of the 26,498 people screened, those with positive anti-HCV were 297(1.1%) (95% CI: 1.00, 1.25). In both sexes anti-HCV positivity increased with age with no significant differences between the sexes (Figure 2). Only nine individuals had HBV and HCV co-infections 

Table 3 could be restricted to the prevalence per age range observed on the current study in Mahama camp while the comparative analysis using the already published RPHIA data could be included in the discussion.

The recent release of the RPHIA final report provides additional details on the age-specific prevalence of HBsAg and anti-HCV, which allowed us to calculate an indirectly standardized prevalence ratio, which we believe is a better approach to comparing the two populations than the original table three. To that end, we have deleted the original table 3 and corresponding results section.

We have additionally altered the text of the Methods section, under statistical methods and data analysis, page 8 -9, lines 151-168 as follows:

To compare the prevalence among refugees in Mahama with the Rwandan general population, we recalculated the prevalence of HBsAg and Anti-HCV within the groups included in the Rwandan Population-based HIV National Impact Assessment (RPHIA), a recent national house-hold survey. recent results from the nationally-representative Rwanda Population-Based HIV Impact Assessment (RPHIA), which published prevalence of hepatitis B and C among Rwandans aged 15-64, we recalculated the crude prevalence of HBsAg and Anti-HCV among 15 to 64 year-old participants in the Mahama screening program among men and women. To create directly comparable estimates that were not confounded by age, we also calculated prevalence that were standardized to the age distribution of RPHIA survey respondents using 5-year age categories. Finally, we reported the indirectly standardized prevalence ratios (SPR) and 95% confidence intervals comparing the observed number of Mahama residents who screened positive for HBsAg or anti-HCV with the number of positive cases that we would have expected if each age group in Mahama had experienced the same age-specific prevalence as reported in the RPHIA report. The interpretation of the SPR is similar to the interpretation of the more commonly used Standardized Mortality Ratio, in that a SPR greater than 1 would reflect that, after adjusting for age, Mahama residents have a higher prevalence than Rwandan general population while a SPR less than 1 reflects a lower prevalence among Mahama residents relative to the general population.

We have added the following text and table to our Results section, page14-15, lines 226-237 :

The prevalence of HBsAg and anti-HCV did not meaningfully change when we restricted our sample to residents aged 15-64 or when we standardized our age distribution to the ages of respondents to the Rwandan Population-based HIV Impact Assessment 2018-2019. Our indirectly standardized prevalence ratios suggest that after adjusting for age, Mahama residents have a 1.86 times higher prevalence of HBsAg than the Rwandan general population (95% CI: 1.74-1.98). The prevalence of HBsAg is, especially elevated among women (SPR=2.44, 95% CI: 2.21-2.69). The prevalence of anti-HCV is significantly lower in the overall Mahama population than among the Rwandan population (SPR=0.84, 95% CI: 0.74-0.96), driven largely by a lower prevalence among men (SPR=0.71, 95% CI: 0.58-0.87). 

Table 3. Prevalence of HBsAg and anti-HCV among Mahama residents aged 15-64 compared to age-standardized prevalence in the general population

 Crude prevalence and 95% CI

among Mahama residents 15-64 Age-standardized prevalence and 95% CI among Mahama residents 15-64 Indirectly standardized prevalence ratio and 95% CI1

HBsAg 

Female 3.1% (2.8%, 3.4%) 3% (2.7%, 3.3%) 2.44 (2.21, 2.69)

Male 4.7% (4.4%, 5.1%) 4.5% (4.2%, 4.9%) 1.58 (1.46, 1.72)

Total 3.9% (3.6%, 4.1%) 3.7% (3.5%, 4%) 1.86 (1.74, 1.98)

Anti-HCV 

Female 1% (0.8%, 1.2%) 1.1% (0.9%, 1.3%) 0.99 (0.83, 1.17)

Male 0.8% (0.7%, 1%) 0.9% (0.7%, 1.1%) 0.71 (0.58, 0.87)

Total 0.9% (0.8%, 1%) 1% (0.9%, 1.1%) 0.84 (0.74, 0.96)

1Data age distribution and age- and sex-specific hepatitis prevalence extracted from the Rwandan Population-based HIV Impact Assessment 2018-2019

Finally, we have relocated any direct comparison to the RPHIA results to the discussion section, page 16-17, lines 255-272 and altered the text to read:

This paper is the first to measure HBsAg and anti-HCV prevalence and assess for associated risk factors among refugee populations living in Rwanda. The 3.8% prevalence of HBsAg shown in our study is slightly higher than what has been reported in Rwanda’s recent population-based survey (2.0%) but comparable to previous estimate from mass screening campaigns among the Rwandan general population (3.9%). This elevated risk among refugees persisted even after we standardized our age distribution to the age of respondents to the RPHIA. Overall, the observed prevalence of HBsAg was almost twice higher among Mahama residents than we would have expected if Mahama residents had exhibited the same age-specific HBsAg prevalence’s as the respondents to the RPHIA survey. As elsewhere in Rwanda, we also found that men are at higher risk than women and that prevalence is highest among middle-aged individuals (16). Anti-HCV prevalence among refugees (1.1%) was similar to the prevalence among Rwandans who participated in the RPHIA (1.2%). However, the observed prevalence of anti-HCV was slightly less among Mahama residents than we would have expected if Mahama residents had exhibited the same age-specific prevalence’s as the respondents to the RPHIA survey. This lower prevalence of anti-HCV among Mahama residents may largely reflect reduced HCV prevalences at older ages. We observed that 2.8% and 3.1% of male and female Mahama residents aged 55-64 were anti-HCV positive, respectively. However, RPHIA reported prevalences of between 4.3% and 11.0% for male and female Rwandans age between 50-64 (6) HCV prevalence among refugees was much lower compared to prevalence from the previously campaign among people aged 25 years and above nationally (6.8%) and in Kirehe district, where Mahama refugee camp is located, (8.4%) (21). Similar to what has been reported in previous campaigns in Rwanda, we found that anti-HCV prevalence increases with age with no differences between the sexes. This increased risk with age may be explained by historic unhygienic medical procedures, especially for those conducted by traditional practitioners, before the establishment infection control policies. However, the risk of anti-HCV among Rwandans 50-64 was also much higher compared to among refugees of the same age. 

Although our campaign in Mahama had extremely high coverage (76.9%) and is likely to be a relatively accurate estimate of the prevalence of HBV and HCV among refugees, we would expect a voluntary screening campaign to result in higher prevalence than a population-based survey as individuals who have prior knowledge or strong reasons to suspect that they may have hepatitis will be more motivated to participate in the campaign. This mechanism could also explain why the previous estimates from Rwanda’s general population mass screening campaigns were higher than those from their recent population-based survey. In general, it appears that the age-specific risk of viral hepatitis in these two populations are generally comparable, with the possible exception of elevated risk of HCV among older Rwandans.

Table 4 : Were patients with viral load at the detection thresholds (20 IU for HBV DNA and 15 IU for HCV RNA) considered as detectable or undetectable? Revise one of the signs of equality (table 4) to avoid confusion.

Comment addressed on Page 16,

Thank you. The sign of equality for undetectable is revised for HBV (˂20 IU/mL) and HCV (˂15IU mL). 

If data are available, it would have been interesting to see proportion of refugees who replicated the most (high viral load e.g., for HBV). Stratifying viral load will help to better orient strategies to limit contamination in the camp.

We have added the requested analysis, as described in the methods section on page 8; lines 147-167, and in the results section on page 15; lines 224-232 along with a table describing this analysis in the appendix

Statistical methods and data analysis: Page 8, lines 148-151

Data were summarized using proportions and 95% confidence intervals (95% CIs). Prevalence were reported overall and stratified by age and sex. To identify risk factors associated with HBsAg and anti-HCV, we dichotomized categorical risk factors and conducted bivariate analysis to calculate crude risk ratios (RR) with 95% confidence intervals and age-and-sex adjusted risk ratios (ARR) using the Mantel-Haenszel method. Adjusted risk ratios were calculated using strata defined by sex and 20-year age intervals. We also assessed associations between having a detectable viral load for HBV DNA or HCV RNA and age, sex, and comorbidities among individuals who screened positive for HBsAg or anti-HCV. As a sensitivity analysis, we also assessed the association between demographic characteristics and having a viral load >20,000 IU/mL, which is the threshold for automatically qualifying for HBV treatment in Rwanda, among patients with detectable HBV DNA. To compare the prevalence among refugees in Mahama with the Rwandan general population, we recalculated the prevalence of HBsAg and Anti-HCV within the groups included in the Rwandan Population-based HIV National Impact Assessment (RPHIA), a recent national house-hold survey. All analyses were conducted using STATA version 15.1 (20).

Results: Page 16, lines 241-247

Table 4 shows that, of the 1006 patients who screened positive for HBsAg, 916 (91.0%) had viral load results returned and of these 916, 781 (85.3%, 95% CI: 82.80, 87.50) had detectable HBV DNA. Male sex and younger age were both associated with viral load positivity for HBV as well as with having a viral load >20,000 UI/mL among those with detectable viral loads (Appendix A). For HCV, of 297 of patients who screened positive, 271 (91.2%) had valid viral load results returned of which 213 (78%, 95% CI: 73.23, 83.33) had detectable HCV RNA positive results. 

Discussion: Paragraph 2:

“We did not observe any association between duration of stay ……transmission within the camp and between the refugees and Rwandan host population is likely rare”

Even if we know that the main transmission route for HBV in Africa is vertical (mother-to-child) and horizontal (during childhood) transmission, it is difficult to conclude that transmission in the camp is rare because this current publication is a cross-sectional study. In addition, other HBV markers have not been evaluated.

The comment is addressed on page 19, lines 297-302

Notably, we did not observe any association between duration of stay in Mahama and risk of HBV or HVC, suggesting that transmission within the camp and between the refugees and the Rwandan host population was is likely rare during our study period. However, it is important to note that this study is cross-sectional study and therefore cannot provide conclusive evidence against transmission within the camp.

Limits:

Since other HBV markers have not been tested (e.g., anti-HBc antibodies), occult hepatitis may escape to screening.

Thank you, the comment is well considered. We have added a sentence in the discussion section to highlight this as a limitation. Page 20, lines 335-337

First, we used a clinical data source, and some missing data, especially on VL results, did occur. Second, risk factors were self-reported and responses may have been affected by social desirability bias. Third, our screening strategy relied on HBsAg, and therefore some cases of occult hepatitis may have escaped detection. 

References

Revise references 6 and 19 (they are not accessible)

Thank you. Reference 6 has been corrected and cited as suggested cited report:

 Ministry of Health. Rwanda population-based HIV impact assessment. Kigali,Rwanda; 2020 Feb.Rwanda Biomedical Centre(RBC). Rwanda Population-Based HIV Impact Assessment (RPHIA) 2018-2019:Final Report [Internet]. Kigali: RBC; 2020 Sep. Available from:

http://phia.icap.columbia.edu

Reference 19 has been deleted, details on VL sample collection and transportation have been added in the method section. Nyirahabihirwe F. Impmentation of a mass screening screening campaign for hepatitis B and C among refugees living in Mahama, Rwanda. Rwinkwavu,Rwanda; 2021.

---

## [Decision Letter · Decision Letter 1]

14 Sep 2021

Prevalence and associated risk factors for hepatitis B and C viruses among refugee populations living in Mahama, Rwanda: A cross-sectional study

PONE-D-21-14910R1

Dear Dr. Kamali,

We’re pleased to inform you that your manuscript has been judged scientifically suitable for publication and will be formally accepted for publication once it meets all outstanding technical requirements.

Kind regards,

Isabelle Chemin, PhD

Academic Editor

PLOS ONE

Additional Editor Comments (optional):

Reviewers' comments:

Reviewer's Responses to Questions

**Comments to the Author**

1. If the authors have adequately addressed your comments raised in a previous round of review and you feel that this manuscript is now acceptable for publication, you may indicate that here to bypass the “Comments to the Author” section, enter your conflict of interest statement in the “Confidential to Editor” section, and submit your "Accept" recommendation.

Reviewer #1: All comments have been addressed

2. Is the manuscript technically sound, and do the data support the conclusions?

Reviewer #1: Yes

3. Has the statistical analysis been performed appropriately and rigorously? 

Reviewer #1: Yes

4. Have the authors made all data underlying the findings in their manuscript fully available?

Reviewer #1: Yes

5. Is the manuscript presented in an intelligible fashion and written in standard English?

Reviewer #1: Yes

6. Review Comments to the Author

Reviewer #1: The authors have given adequate answers to questions and addressed all comments raised in a previous round of review.

Remark : The storage of RNA at 4 degree even for less than 72 hours is not optimum and can lead to under estimation in those samples.

7. PLOS authors have the option to publish the peer review history of their article (what does this mean?). If published, this will include your full peer review and any attached files.

Reviewer #1: No

---

## [Editor Report · Acceptance letter]

22 Sep 2021

PONE-D-21-14910R1 

Prevalence and associated risk factors for hepatitis B and C viruses among refugee populations living in Mahama, Rwanda: A cross-sectional study 

Dear Dr. Kamali:

I'm pleased to inform you that your manuscript has been deemed suitable for publication in PLOS ONE. Congratulations! Your manuscript is now with our production department. 

Kind regards, 

on behalf of

Mrs Isabelle Chemin 

Academic Editor

PLOS ONE